



# Joint analysis of the magnetic field and Total Gradient Intensity in Central Europe

Maurizio Milano [1], Maurizio Fedi [2], J. Derek Fairhead [3]

[1] Centro Interdipartimentale di Ricerca L.U.P.T. – Università degli Studi di Napoli Federico II.
[2] Dipartimento di Scienze della Terra, dell'Ambiente e delle Risorse - Università degli Studi di Napoli Federico II.
[3] School of Earth and Environment, University of Leeds, UK.

*Correspondence to*: Maurizio Milano (maurizio.milano@unina.it)

**Abstract.** In the European region, the magnetic field at satellite altitudes (~350 km) is mainly defined by a long-wavelength magnetic-low called here the Central Europe Magnetic Low (CEML), located to the southwest of the Trans European Suture Zone (TESZ).

We studied this area by a joint analysis of the magnetic and total gradient ($\nabla T$) anomaly maps, for a range of different altitudes of 5 km, 100 km and 350 km. Tests on synthetic models showed the usefulness of the joint analysis at various altitudes to identify reverse dipolar anomalies and to characterize areas in which magnetization is weak. By this way we identified areas where either reversely or normally magnetized sources are locally dominant. At a European scale these anomalies are sparse, with a low degree of coalescence effect. The $\nabla T$ map indeed presents generally small values within the CEML area, indicating that the Palaeozoic Platform is weakly magnetized.

At 350 km altitude, the TESZ effect is largely dominant: with intense $\nabla T$ highs above the East European Craton (EEC) and very small values above the Palaeozoic Platform, this again denoting a weakly magnetized crust. Small coalescence effects are masked by the trend of the TESZ.

Although we identified sparsely located reversely magnetized sources in the Palaeozoic Platform of the CEML, the joint analysis does not support a model of a generally reversely magnetized crust. Instead, our analysis strongly favors the hypothesis that the CEML anomaly is mainly caused by a sharp contrast between the magnetic properties of EEC and Palaeozoic Platform.

## 1 Introduction

In the last decades, potential field satellite observations have become fundamental in studying the large-scale component of the gravity and magnetic field originating in the Earth's lithosphere. Since the first satellite derived maps of the European magnetic field were constructed in the 1980s, many researchers have tried to give a reasonable explanation to the extended magnetic low covering most of central Europe (e.g. Ravat et al., 1993; Taylor and Ravat, 1995). This crustal magnetic anomaly is clearly visible on the satellite map MF7 (Maus, 2010) (Fig. 1), based on crustal magnetic field measurements at 350 km altitude collected from the CHAMP satellite mission. At a first sight (Fig. 1) the Central European Magnetic Low (CEML) seems to be part of a very large reverse-polarity magnetic anomaly, with its positive located to the northeast of the Trans



European Suture Zone (TESZ), and the weaker low amplitude negative CEML to the southwest. Previous geological interpretations of the CEML were carried out combining petrologic and geological studies with satellite magnetic field modelling (e.g. Ravat et al., 1993; Taylor and Ravat, 1995; Pucher and Wonik, 1998). They considered the observed magnetic anomaly as a coalescence effect occurring at satellite altitude, due to the widespread concentration of magnetic sources with

5 strong reverse magnetization in the upper crust of central Europe. Based on the satellite data alone, the proposed reverse-polarity of the TESZ magnetic anomaly is not totally resolved or explained. Recently, Milano et al. (2016) performed a multiscale analysis on a three-dimensional aeromagnetic dataset in central Europe and produced a model of the deep magnetic sources; they found that the origin of the high-altitude magnetic field in central Europe was mainly due to changes of crustal thickness and of physical properties between western and eastern Europe. To resolve the situation we have both enhanced the

10 spectral content of magnetic data and applied new analytical methods to the data. These are:

A)        The crustal satellite derived magnetic field MF7 has a limited spectral content with $\lambda > 350$ km. The spectral content has been significantly improved by having recent access to the terrestrial magnetic data EMMP (Fletcher et al 2011) gridded a 1 km allowing upward continued datasets at 5 km and 100 km to be generated. The satellite dataset MF7 is preferred at 350 km altitude due to potential edge effects of the former.

B)        Application of the Total Gradient ($\nabla T$) of the magnetic field. Here, the dipolar shape of the magnetic anomalies is

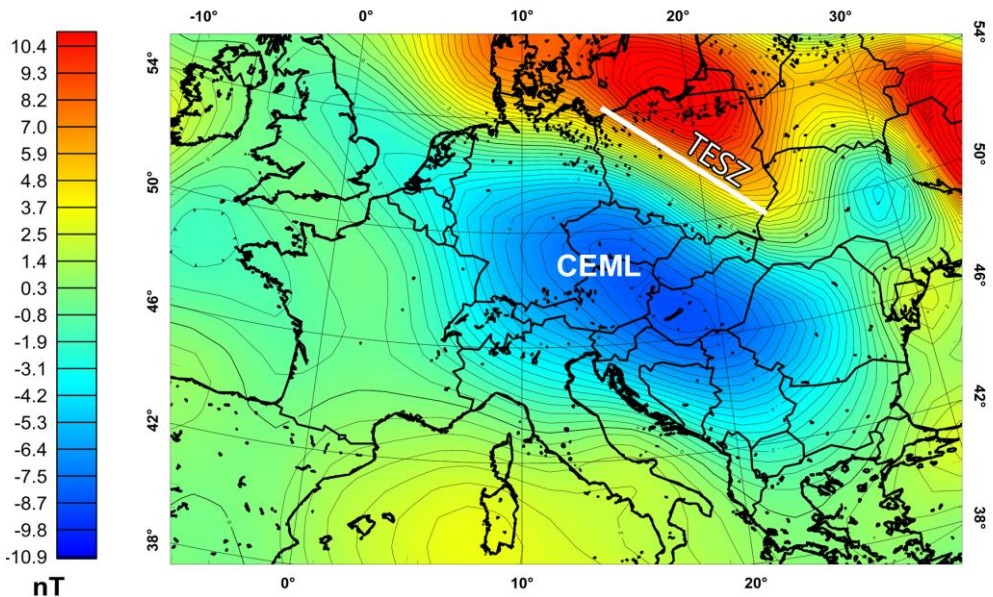

**Figure 1. MF7 magnetic field model at 350 km altitude. (CEML: Central European Magnetic Low; TESZ: Trans European Suture Zone).**

replaced by highs which tend to be placed directly over the source location (Nabighian, 1972). This provides a clearer image



of the magnetic field over Central Europe and better identify the location of the magnetic sources. Such kind of analysis will help to evaluate and thus understand how the mechanism of coalescence is related to the CEML anomaly.

## 2 The Geological Setting of the European Crust

The structural framework of the European continent is the result of several geodynamic events which produced a complex 'collage' of crustal blocks differing in geological and geophysical properties (Fig. 2). Central Europe formed while the supercontinent Rodinia dismantled from Neoproterozoic events (McCann, 2008), yielding the tectonic structures of the Alpine system, the Carpathians chain, the Pannonian basin, the Bohemian massif and the Trans European Suture Zone (TESZ). TESZ is the main crustal boundary in central Europe as a result of the movement and juxtaposition between the stable and olden north-eastern East European Craton (EEC), or Precambrian platform, and the younger south-western Palaeozoic platform (PP), during the Caledonian and Variscan orogenic events (e.g. Pharaoh, 1999; Thybo, 2001; Banka, 2002). TESZ includes two

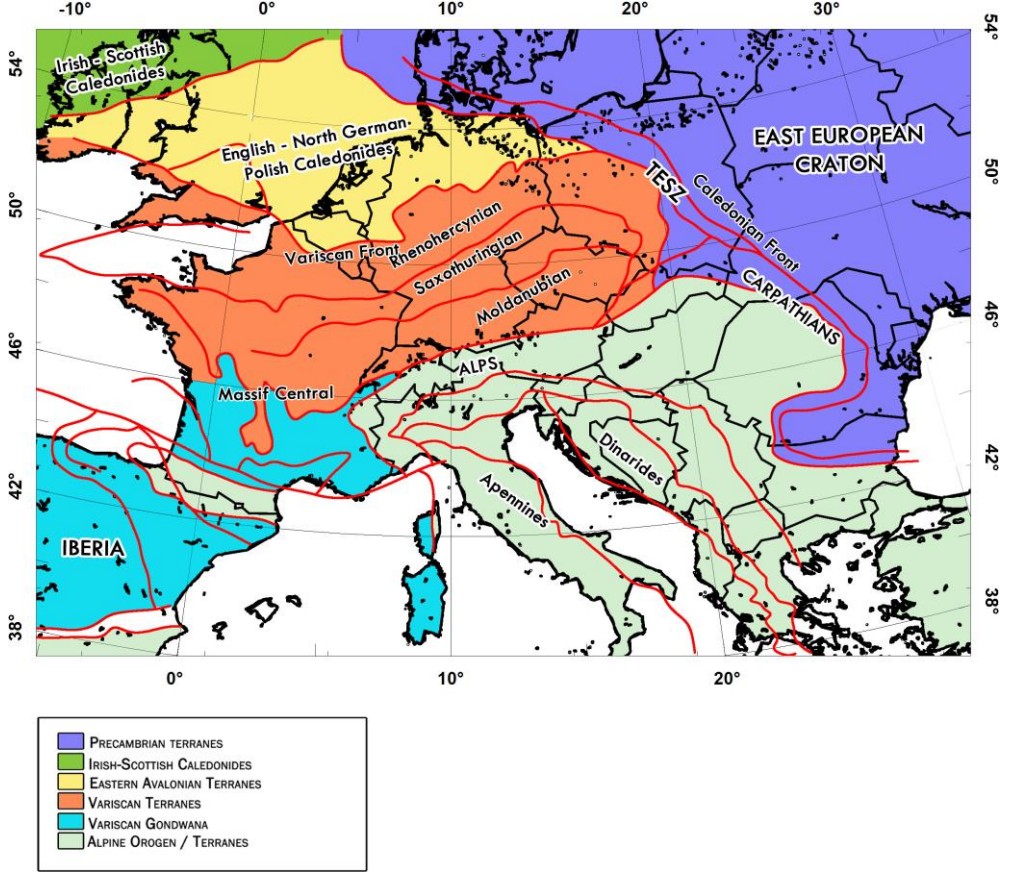

**Figure 2. Tectonic setting of the European continent (redrawn after Blundell et al., 1992, and Plant et al., 1998).**



main structural segments: the Sorgenfrei–Tornquist Zone (STZ) in the northwestern part and the Teisseyre–Tornquist Zone (TTZ) in the southeastern part. It extends from the central North Sea via the Baltic Sea (northwest) to the Black Sea (southeast) (Pharaoh, 1999). In northeastern Europe, the Baltica paleocontinent evolved during several Precambrian episodes of crustal accretion and reworking. It comprises the Fennoscandian and Ukrainian shields and the East European Craton. EEC extends from Denmark into Russia on the eastern side and includes the segments of Sarmatia, Volgo-Uralia and Fennoscandia, which differ in forming dynamics, lithological and tectonic features (Bogdanova et al., 2006). Generally, the EEC is characterized by a thick and cold crust composed mainly by Precambrian rocks evolved during

The structural setting of the European crust has been modelled thanks to the large amounts of seismic and potential field data which allowed to investigate the lithosphere down to 60 km depth (e.g., Guterch et al., 1999; Pharaoh, 1999; Mushayandebvu et al., 2001; Banka et al., 2002; Majorowicz and Wybraniec, 2011). Several seismic reflection surveys, such as BIRPS (UK), CROP (ltaly), DEKORP (Germany), ECORS (France) NFP 20 (Switzerland) and the European Geotraverse (EGT) project aimed at developing a three-dimensional representation of the geological structures and properties of the European lithosphere (Freeman and Mueller, 1990; Blundell et al., 1992). Moreover, further deep seismic sounding (DSS) studies were carried out in central Europe, such as KTB in Germany (DEKORP Research Group 1988; Gebrande et al. 1989) and the POLONAISE'97, CELEBRATION 2000, ALP 2002 and SUDETES 2003 (Guterch et al., 1999, 2003a; Grad et al., 2003a; Brückl et al., 2003) in Poland, revealing the strong contrast between the PP and the EEC and the structural features of the TESZ area. Seismic data interpretation revealed the cratonic part of Europe characterized by anomalous thickness, where the roots may reach the maximum depth of 55-60 km (Artemieva and Meissner, 2012), as in the Ukrainian shield, Uralides and Baltic shield. The Variscan terranes in central Europe, instead, show a uniform crust thickness around 28-32 km (e.g. EUROBRIDGE SWG, 1999; Korja and Heikkinen, 2005; Guterch and Grad, 2006). These variations in crustal thicknesses are shown in Figure 3a using the Moho depth map of Europe produced by Grad et al. (2009). A clear depth variation is observed in the TESZ area, passing from the thin Palaeozoic platform to the EEC, with a 50 km increase of crustal thickness in southern Sweden, Denmark, Baltic Sea, Poland and Slovakia (BABEL Working Group, 1993a; Giese and Pavlenkova, 1988; Thybo, 1990, 2001; Guterch and Grad., 2006).

Deep-borehole temperature measurements also reveal a strong difference in geothermal properties between western and eastern Europe (e.g. Čermák et al., 1989; Hurting et al., 1992; Plewa, 1998; Królikowski, 2006) as illustrated by the heat flow map (Fig. 3b). Low regional heat flow values (30-40 mW m$^{-2}$) were measured within the Baltic shield (Precambrian platform) which is typical of a thick lithosphere, whereas relatively high heat flow values were observed in specific areas of depressions of the EEC basement (e.g. Čermák and Rybach, 1979). Central Europe, instead, is dominated mainly by extended high heat flow zones trending from east to the west. boundaries, allowing considering an upper-mantle contribution to the geomagnetic field (Chiozzi et al., 2005).

As regards rock magnetism, reverse magnetization in central Europe was evidenced by petrological analysis of Permian quartz porphyries beneath the North German sedimentary basin, containing pyrrhotite-metasedimentary rocks with susceptibility values up to 2 x10$^{-3}$ cgs and Koenigsberger ratio (Q) of about 100 (Henkel, 1994). Further studies of Palaeozoic metamorphic





rocks revealed reversely magnetized sources in western Germany (Pucher, 1994) and additional analysis of metasedimentary rocks showed a possible overprinting during the Permo-Carboniferous reverse superchron (Thominski et al., 1993).

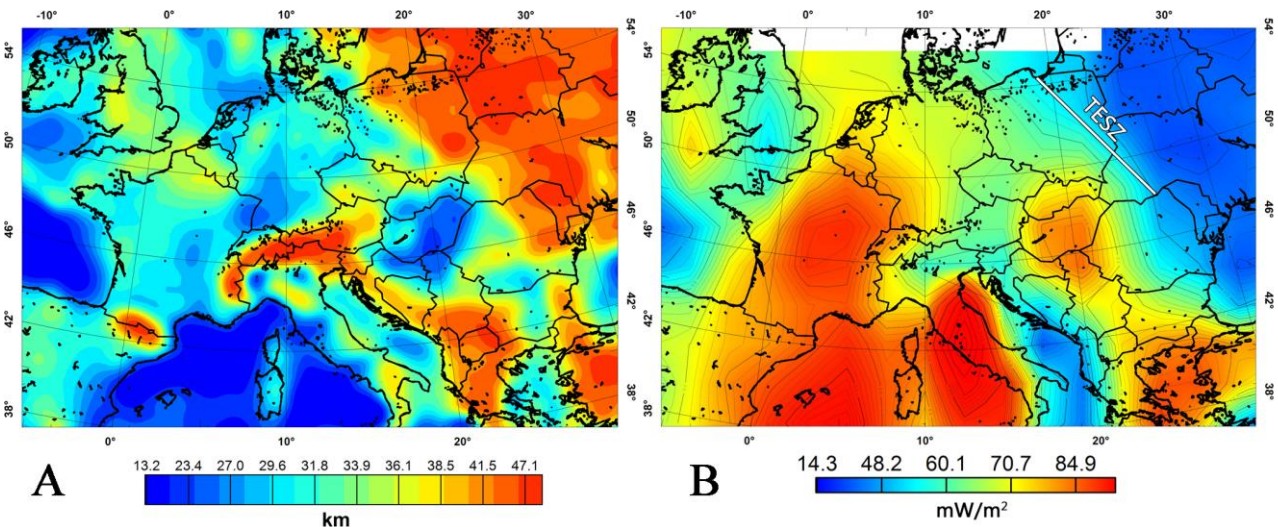

**Figure 3. a) Depth to the Moho beneath the European continent (data from Grad et al., 2009). b) Heat flow map of Europe (data from Davies, 2013).**

## 3 The Terrestrial Magnetic Field Compilation of Europe

Satellite magnetic field model (e.g MF7) has been used successfully in defining the crustal magnetic field for wavelengths λ
> 350 km, however, it is unable to resolve anomalies at shorter wavelength due to high altitude of satellite measurements. To investigate the nature and cause of the anomaly field associated with the TESZ and CEML, terrestrial magnetic data is needed. This study has had access to the European and Mediterranean Magnetic Project (EMMP) dataset, from Getech (Fletcher et al., 2011), where all the available magnetic data of the European countries (including ground, air and marine data) have been digitized, reprocessed and merged to produce a high-resolution 1 km grid at 1 km above topography of total magnetic intensity
(TMI) field (see Fletcher et al., 2011 for more details). As indicated in the introduction, CEML has been previously interpreted on the base of satellite-altitude magnetic field data as the result of coalescence between small-scale magnetic anomalies, with mainly reverse polarization. Now having data at different altitudes and resolutions by upward continuing the terrestrial compilation data, our analysis can be performed at different altitudes which allows us to recognize the contributes of the various magnetic sources to the CEML at different scales. Regarding the scale of analysis, we decided to analyze the magnetic
field map at 100 km and 350 km altitude. The maps at these altitudes were produced by the upward continuation technique, a standard procedure to transform the field data measured at a given altitude to a higher level. The transformation consists of a convolution, which may be performed in the Fourier domain (Baranov, 1975; Fedi et al., 2012). We first filled small areas of





no data by a Maximum Entropy interpolation (Gibert and Galdeano, 1985) and, then, performed a draped-to-level upward continuation to a 5 km altitude level using the wavelet-based algorithm by Ridsdill Smith (2000) and the Digital Elevation Model SRTM30 (Farr et al., 2007) (Fig. 4a). Then, we performed a level-to-level upward continuation using the classical convolution integral:

$$U(x,y,z) = \frac{1}{2\pi} \int_S U(\xi,\eta,0) \frac{z}{[(x-\xi)^2 + (y-\eta)^2 + z^2]^{3/2}} \, d\xi d\eta \tag{1}$$

where $U(x,y,z)$ is the potential field upward continued to the scale $z$, $U(\xi,\eta,0)$ is the potential field at level $z=0$ and $z/[(x-\xi)^2+(y-\eta)^2+z^2]^{3/2}$ is the upward continuation operator. In Figure 4b we show the map of the upward continuation magnetic field at 100 km and 350 km altitude. Note that, in principle, equation 1 is applicable to fields known continuously on the entire flat surface. Our data, instead, are discrete datasets, containing equally-spaced data distributed on a finite surface. However, as shown in Fedi et al. (2009), accurate upward continuation may be computed if the data are known on a surface larger than the area of

interest and/or by extrapolating data beyond its borders with a suitable algorithm. Upward continuation of data, indeed, may produce a border (or edge) effect near the grid margins due to space-domain aliasing (Oppenheim and Schafer, 1975; Fedi et al. 2012; Mastellone et al., 2014). To reduce this effect, upward continuation has been performed by extrapolating the aeromagnetic dataset onto a larger area using a Maximum Entropy algorithm.

Furthermore, it may be useful to calculate the extent $\Delta L$ of the upward continuation operator $z/[(x-\xi)^2+(y-\eta)^2+z^2]^{3/2}$ in order

to estimate the portion of the map which may be mostly affected by the border effect. We may simply map the operator at different altitudes and evaluate $\Delta L$ from its half-width. We define $\Delta L$ as the distance occurring from the maximum to the point where it is decreased of a factor $1/e$, $e$ being the Neper number. Thus, we estimated $\Delta L= 96$ km at 100 km altitude, which means that, using $\Delta L$, we may be rather confident in the upward continued field computed at 100 km altitude over most of the European continent, excluding a border defined by $\Delta L$. At 350 km altitude, instead, the border effect is significantly greater

with $\Delta L= 347$ km as well as the discrepancy with the MF7 model map (Fig. 4c). Therefore, the total gradient analysis at satellite altitude is performed on the map of the MF7 magnetic data at 350 km (Fig. 4d). The European magnetic field map at 5 km altitude is mostly characterized by short-wavelength anomalies. The first and most important feature of the map is the visible contrast in the distribution and intensity of the anomalies located over the Palaeozoic and EEC regions. As revealed by seismic modelling, the TESZ is here a sharp geological/tectonic boundary, which controls the main regional trend of the magnetic

field. It suggests a very sharp change in the crust's magnetic properties, in addition to geological, compositional and geothermal differences. The anomalies associated with the TESZ define the most important magnetic anomaly zone of central Europe and relate to sources involving the whole crust. The TESZ magnetic anomaly is especially visible in Poland and Romania and it divides these countries in two opposite magnetic provinces: the south-western area, almost covered by magnetic lows, and the north-eastern region, characterized by numerous magnetic highs. Actually, this anomaly trend may be extended all over the

European continent, identifying the TESZ as the margin between these two opposite magnetic behaviours (Banka et al., 2002; Williamson et al., 2002; Fairhead, 2015). Such magnetic style is imaged even more clearly at 100 km altitude, where the map shows a considerably smoothed magnetic field when compared to that at 5 km. Here, the strong difference between the EEC



and Palaeozoic crusts is well appreciable by the development of an extended magnetic low in central Europe in contrast with
an area of magnetic highs to the NE, anticipating the formation of the reverse dipolar anomaly above the TESZ structure and
the CEML at 350 km altitude (Fig. 4c). Following Ravat et al. (1993) the origin of the CEML could be associated to the
considerable difference in crustal thickness and heat flow between the EEC and Palaeozoic platforms. Taylor and Ravat (1995),

5    however, carried out an alternative interpretation based on the predominance of remanent magnetization within the Palaeozoic
crust. Their sketched model consists of two prismatic blocks representing, respectively, the reversely magnetized Palaeozoic



**Figure 4. Aeromagnetic field map of Europe at 5 km (a), 100 km (b), 350 km altitude (c); Map of MF7 magnetic field model
at 350 km altitude (d).**

platform and the normally magnetized Precambrian crust. Therefore, the main origin of the CEML was assigned to a pattern
of relatively small reversely magnetized sources in the upper-middle crust of the Palaeozoic platform, coalescing at satellite
altitude and so forming the extended long-wavelength magnetic low. Pucher and Wonik (1998) proposed a further
interpretation still based on reversely-magnetized sources into the Paleozoic crust, based solely to the magnetic properties of

10   central Europe, while the EEC was not considered contributing to the CEML formation. The effect of coalescence between





short-scale anomalies, shared by these magnetic models of Central Europe, could be verified if we compare simultaneously the field map at small and large altitudes.

## 4 Total gradient analysis

The dipolar shape of the magnetic field anomalies represents a troublesome limitation in the study and identification of the magnetic sources, due to the non-vertical direction of total magnetization and inducing field. This issue may be overcome by employing methods able to remove almost completely the dipolar behavior of the field and so providing maps of anomalies more directly related to the source position. One such method is the total gradient technique.

The total gradient modulus of the magnetic field $T$ ($|\nabla T|$) is defined as:

$$| \nabla T |= \sqrt{\left( \frac{\partial T}{\partial x} \right)^2 + \left( \frac{\partial T}{\partial y} \right)^2 + \left( \frac{\partial T}{\partial z} \right)^2} \qquad (2)$$

where $\partial T/\partial x$, $\partial T/\partial y$ and $\partial T/\partial z$ are the partial derivatives of the total magnetic field ($T$) with respect to the directions $x$, $y$, and $z$.

The total gradient anomalies have two important features:

i) they are located directly over the source bodies and may help localizing the sources edges (e.g. Roest et al., 1992; MacLeod, 1993; Roest and Pilkington, 1993; Salem et al., 2002; Shearer and Li, 2004; Paoletti et al., 2016);

ii) they are positive anomalies, no matter whether the magnetization is direct or reverse. In fact, $|\nabla T|$ is completely independent of the directions of both the inducing field and total magnetization in the 2D case (Nabighian, 1972) and only weakly dependent on the directions of $T$ in the 3D case (Haney et al., 2003). Salem et al., (2002) pointed out that, in the 3D case, a maximum shift for I of 30° does not cause appreciable changes in the total gradient amplitudes.

In this paper we specifically focus our analysis on the total gradient anomalies to identify the presence of sources with strong remanent component of magnetization and to interpret whether large-scale magnetic anomalies can be produced by coalescence effect. To clarify the problem, we consider a synthetic model composed of three magnetic sources: B1 and B2, with normal magnetization ($I=I_F=65°$, $D_F=3°$), and B3, reversely magnetized with $I=-15°$ and $D=230°$ (Fig. 5), where $I_F$ and $D_F$ indicate the direction of the inducing field, while $I$, $D$ indicate those of the total magnetization. These parameters where chosen according to the results of paleomagnetic analysis in central Europe carried out by Pucher (1994) and which were used to interpret the regional magnetic field over Europe by Pucher and Wonik (1998). The map of the magnetic field ($T$) (Fig. 5a) shows two dipolar anomalies B1 and B2 with their magnetic lows and highs located to the NE and SW, respectively. The source B3, instead, is not well identified, since its magnetic effect is covered by the anomalies produced by B1 and B2. The presence of an anomaly due to B3 could be perhaps identified by the slight prolongation of the magnetic high of B2 towards the location of B3 and a by weak increase in intensity of the magnetic low of B1 to the SW. This example shows that the





presence of a reversely magnetized source into a complex magnetic environment is hard to be identified and isolated from the map of the magnetic field. The total gradient intensity map of the magnetic field (Fig. 5b) transforms the dipolar shape of the magnetic anomalies into high amplitude monopole shape anomalies which now provide detailed information on the source locations. The reversely magnetized source B3 is now well imaged and clearly distinguished from the other sources. Note that

in the map of the magnetic field the field lows (blue color) stand for the lows of the dipolar magnetic anomalies, while in the total gradient intensity map the smallest values are around zero and, most important, denote the areas in which magnetization is very weak. So, the total gradient intensity highs yield an easy and effective tool to indicate, first of all, the relative presence or absence of magnetized sources and, secondly, the location of sources, even in case of strong remanent magnetization. We will use both these properties for analyzing the magnetic field in Europe.

As previously indicated, the two main hypotheses put forward to explain the CEML anomaly are:

A) a pattern of reversely magnetized sources in the upper-middle crust of the Paleozoic platform, coalescing at satellite altitude;

B) a geological contact between a thick magnetized crust (EEC) and a thin low susceptibility crust (Central Europe) with high heat flow.

In order to evaluate these hypotheses, we will now consider two synthetic cases. The first is related to the hypothesis A). To this end, we produced a new model with 19 prismatic sources to simulate the magnetic crust of central Europe in a reversely-magnetized environment. The magnetization intensity ranged between 1.5 and 2.5 A/m and the direction of magnetization was the same as that of the reversely magnetized body B3 in Figure 5. The magnetic field and total gradient intensity map were calculated at 1 km, 100 km and 350 km altitude (Fig. 6), to better compare the results with the total gradient intensity maps of

Europe. At small altitudes, the short-wavelength anomalies clearly identify the location of the sources both in the $T$ and $|\nabla T|$ maps. At 100 km altitude the magnetic field is considerably smoothed, and the coalescence between the magnetic anomalies begins to be relevant. However, at such altitude is extremely difficult to identify the source location from the $T$ map.

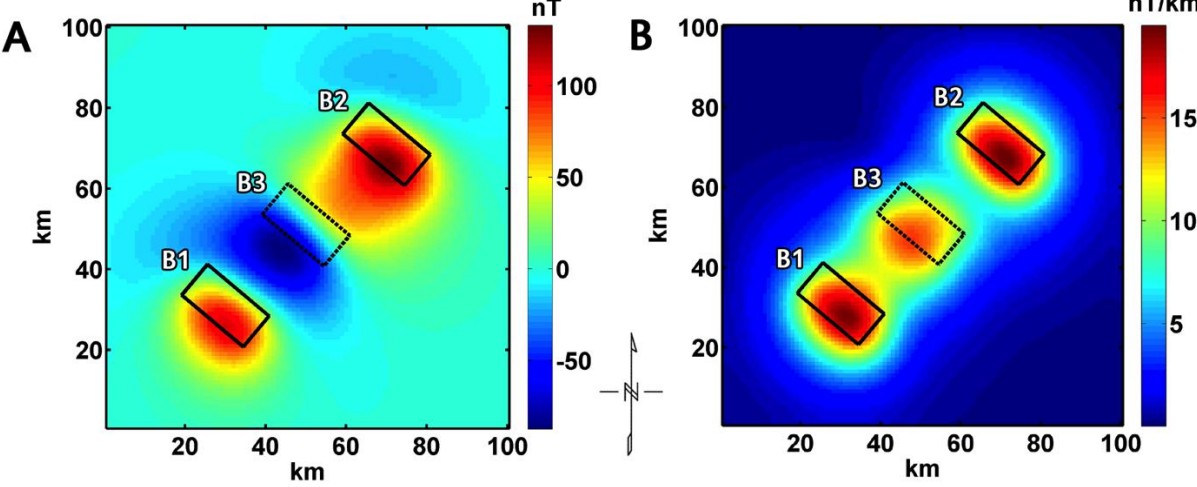

Figure 5. Multi-source model with B1 and B2 normaly magnetized and B3 reversely magnetized. a) $T$ map; b) $|\nabla T|$ map.





On the other hand, the total gradient intensity map allows recognizing the location of the magnetic sources, especially those with higher magnetization that produce stronger anomalies at large altitudes. Note that, similarly to the previous model, the magnetic lows in $T$ are mostly substituted by peaks of $|\nabla T|$ highs, so demonstrating again the ability of the total gradient technique to estimate the presence of reversely magnetized sources.

5    In Figure 6c the magnetic field at 350 km altitude is characterized by a large reverse dipolar anomaly. The intensity of such magnetic anomaly is very weak, meaning that shallow sources has very low contribution at satellite-altitudes, which could be

**Figure 6. Synthetic model of the multiple shallow sources reversely magnetized. The magnetic field ($T$) and total gradient $|\nabla T|$ maps are shown at 1 km (a), 100 km (b) and 350 km (c) altitudes.**

completely negligible in presence of additional deeper sources. Here the total gradient intensity map shows a large amplitude

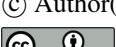


high located toward the magnetic low in the south-western region and suggest the presence of highly magnetized bodies. These results show that, in case of coalescence effect between multiple shallow sources, we can even identify and discriminate the source location by evaluating the total gradient intensity maps, regardless of the direction of magnetization. Therefore, taking advantage of these synthetic model results, we are confident of the presence of reverse magnetization and the coalescence

effect in the European magnetic field maps.

The second hypothesis (B) is analyzed by building a new synthetic model of the magnetic crust in order to simulate the magnetic anomaly above the TESZ. We consider two adjacent prismatic bodies representing the Paleozoic and the EEC with crustal thickness of 30 km and 40 km, according to a seismic model (e.g. Guterch and Grad, 2006), and induced-only magnetization of 1 A/m and 2 A/m, respectively. Thus, we assume no remanent magnetization for both the sources. The

magnetic field and the total gradient intensity calculated at 350 km altitude are shown in Figure 7. The map of the magnetic field shows a reverse dipolar magnetic anomaly with a magnetic high above the synthetic red body (Precambrian crust) and a related magnetic low to the SW, say above the Paleozoic platform (blue body) (Fig. 7a). The $|\nabla T|$ map, instead, has highs located roughly along the contact surface (Fig. 7b) and a wide gradient zone extending toward central Europe. This simple synthetic test clearly points out that the contact between two crustal platforms differing in magnetization and thickness could

explain the large-scale magnetic anomaly above central Europe and its reverse dipolar shape. Obviously intermediate models, considering the magnetic features of both the models A and B can occur. In the next section we will analyze the field and its total gradient intensity taking into account the results of our synthetic source distributions.

## 5 Interpretation of the CEML

We computed the total gradient intensity maps of Europe, by upward continuation of the aeromagnetic dataset of Europe

from 5 km to 100 km and of the MF7 model-data at 350 km altitude. As discussed previously the simultaneous analysis of the total gradient intensity maps at different altitudes allows evaluating in detail the magnetic properties of the area, because it can effectively outline: a) the relative presence or absence of magnetized sources; b) the location of sources with strong remanent component of magnetization; c) the occurrence of coalescence at large altitude. The interpretation of the regional magnetic field is indeed improved if the contributions of the shallow sources can be identified in the low-altitude map. We show the

maps of the total gradient intensity at 5 km, 100 km and 350 km altitude, in Figure 8 a, b and c, respectively. The lineaments of the main tectonic structures (from Blundell et al., 1992) are shown with white lines.





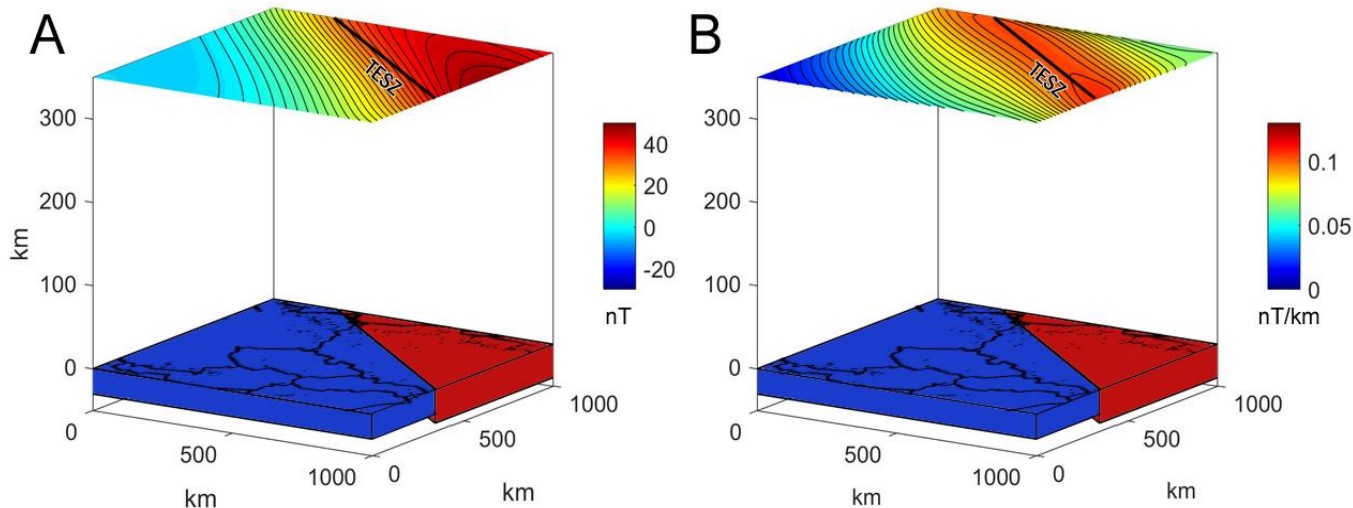

**Figure 7. Contact-like 3D model of central Europe and related magnetic field (a) and total gradient intensity (b), at 350 km altitude. This model refers to the hypothesis B described in this paper, where the main magnetic source is a magnetic contact between the Paleozoic platform (blue body) and the Precambrian platform (red body).**

## 5.1 Magnetic Field at 5 km.

The low-altitude $|\nabla T|$ map gives an interesting representation of the main magnetic provinces of the European crust. The north-eastern part of Europe, including the East European Craton (EEC) and the thick crust of the Baltic shield, is mostly defined by

short-wavelength and well-defined anomalies. The intense distribution of the $|\nabla T|$ highs suggests a strong magnetization of the Precambrian upper-crust. The low detail of the total gradient intensity above the EEC, however, is a mere consequence of our colormap choice of saturating anomalies exceeding 15 nT/km. In central Europe, $|\nabla T|$ highs above Poland highlight the different granitoid massifs of Pomerania, Mazovia and Dobrzyn (Grabowska and Bojdys, 2001; McCann, 2008) and the structural lineaments with WNW–ESE trend of the Polish Lithuanian terranes and NNE trend of the East Lithuanian belt

structures, while rounded and well-defined anomalies above the OMB (Osnitsk–Mikashevichi Igneous Belt) are associated to batholiths and diorites/gabbros intrusions (Bogdanova et al., 2006). In northern Europe, total gradient anomalies are associated to the Caledonian basement and to deep relics of the Scandinavian crust, formed during the collision between Baltica and Avalonia. In central Variscan Europe, instead, several $|\nabla T|$ highs are well displayed over Germany, marking the boundaries of the Caledonian basement to the NW, the Rheno-Hercynian zone in the central area and the Moldanubian zone and Alpine

system to the SE (Gabriel et al., 2011). According to Dallmeyer et al. (1995) the origin of the magnetic anomaly distribution over central Europe may be assigned to sources occurring in the pre-Variscan basement, to Variscan and late-Variscan mafic intrusions and extrusions, and to Cenozoic volcanic activities. $|\nabla T|$ highs define also the magnetic sources forming the Mid German Crystalline Rise (MGCR), which was firstly interpreted as a magmatic arc, formed above a south-dipping subduction



zone in Upper Devonian and Carboniferous times, and then reinterpreted by Oncken (1997) using a more complex model based on two different crusts involved in geodynamic events during Carboniferous. Above the Saxothuringian unit (ST) the total gradient anomalies are associated to several granitoids and metamorphic rocks, which are distributed in two different ranges of depth, about 2 and 11 km, as revealed by spectral analysis (Bosum and Wonik, 1991). Such magnetic environment

marks a clear difference from the adjacent and weakly magnetized Rhenohercynian and Moldanubian zones. However, in the Rhenohercynian unit we find also total gradient intensity highs located above young magmatic regions (Gabriel et al., 2011), such as the Soest magnetic anomalies, characterized by sets of reversely magnetized sources. In the Moldanubian zone some magnetic evidences occur southwards to the MD margin, coinciding with the Donau Line, a prolongation of the northern part of the Massif Central (Bosum and Wonik, 1991). These magnetic anomalies are probably associated to basic and ultrabasic

rocks, which could extend greatly with depth (Franke, 1989). In the eastern parts of Germany and the Czech Republic total gradient intensity highs are located over the Bohemian Massif (BM), representing the eastern side of the Variscan belt, a Paleozoic chain extended from the Iberic to Bohemian complexes in Central Europe (Tomek et al., 1997). This area is characterized by an anomalous crustal thickness with respect to the average thickness of the Paleozoic platform (~35 km). In the north-western sector of the Bohemian massif, the Eger Graben (EG) and its surroundings are defined by isolated magnetic

anomalies probably generated by young volcanism reactivating the Saxothuringian-Moldanubian suture zone (Dallmeyer et al., 1995). Previous magnetic studies of the BM pointed out that this region might be considered as a 'microplate', highly different from the adjacent territories (Bosum and Wonik, 1991). Most of the total gradient anomalies are correlated with the outcropping Cambro-Ordovician basement, the crystalline basement and, in the eastern side, with young magmatic structures related to recent tectonic movements. In western Europe, strong $|\nabla T|$ highs occur above the Midland Microcraton (MM) in

Southern England and continue down to the North Sea and Brabant Massif (BM), showing a very different magnetic environment with respect to the close Rhenohercynian Zone to the East. The magnetic anomalies of MM are mainly related to the outcrop of the Neoproterozoic basement and to local Carboniferous intrusions (Banka et al., 2002; Pharaoh and Gibbson, 1994). In Eastern England patterns of small magnetic anomalies are associated to calcalkaline magmatic rocks and to the southern metasedimentary rocks of the Brabant Massif in northern Belgium (Pharaoh et al., 1993; De Vos et al., 1993). The

Paris Basin magnetic anomaly (PB) is the main magnetic feature of western Europe, extending for 400 km from central France to the English Channel with a S-NE linear trending. Its interpretation is still not completely solved, but the origin may be attributed to ophiolitic fragments carried up to the shallow crust by Late Variscan faulting episodes (Autran et al., 1992). In southern Europe strong $|\nabla T|$ highs occur above the Alpine system, associated with ophiolitic rocks and with a broad distortion of the lower crust, above the continental margins of the Tyrrhenian Sea, related to crustal fractures of the deep geodynamic

events (Rehault et al., 1987), and the Balkan region and the Carpathians mountain chain, where short-wavelength anomalies are mainly associated to magmatic rocks formed during several volcanic events of different ages, while smoothed anomalies extended to the North have origin into the magnetic basement beneath the Bulgarian foreland (Trifonova et al., 2009). In the Romanian region, the main magnetic contribution is associated to the Neogene-Quaternary volcanism beneath the Pannonian basin (Boccaletti et al., 1976), a back-arc basin formed during the complex evolution of the Carpathians-Alpine system





(Horváth, 1993). Following Kis et al. (2011), in the Pannonian basin xenolith and peridotite rocks formed in the deep crust-upper mantle and were carried up into the shallow crust.







**Figure 8. Total gradient intensity maps of the aeromagnetic dataset at 5 km (a) and 100 km altitude (b); total gradient map of the MF7 magnetic field model at 350 km altitude (c). White lines represent the main tectonic structures (from Blundell et al., 1992) ABM: Anglo-Brabant Massif; AMA: Adriatic magnetic anomaly; AS: Alpine System; BM: Bohemian Massif; MC: Massif Central; MD: Moldanubian zone; MGCH: Mid German Crystalline High; MM: Midland Microcraton; OMB: Osnitsk–Mikashevichi Igneous Belt; PB: Pannonian basin.**

## 5.2 Magnetic Field at 100 km.

At 100 km altitude (Fig. 8b) we observe mainly long-wavelength anomalies, whereas small-scale anomalies tend to vanish or to coalesce. Here, a strong difference between the EEC and Paleozoic platforms occurs clearly, because of the intense $|\nabla T|$ highs covering most of the EEC and, on the other hand, the substantially non-magnetic behavior of the Paleozoic platform, characterized by few sparse, weak and isolated anomalies. Huge $|\nabla T|$ highs are located above the TESZ region which, according to Milano et al. (2016), suggests the TESZ as a deep magnetic structure involving the whole crust down to the Moho boundary. This picture seems so favor the hypothesis B) as the main cause of the CEML. In fact, the $|\nabla T|$ map shows clearly that the weak and sparse nature of the magnetization in central Europe is not able to characterize, through the mechanism of coalescence, the anomaly field at large altitudes.

However, a coalescence effect can supply additional contribution to the large-scale magnetic field. Apart the TESZ, the most noticeable $|\nabla T|$ high is observed above the Adriatic region, well recognizable also in the magnetic field map as the Adriatic Magnetic Anomaly (AMA). However, in central Europe we can distinguish some slight and isolated $|\nabla T|$ highs especially above Eastern Germany, central Czech Republic and south-Eastern Europe. Rounded and separated highs are also recognized in southern England and Belgium. Comparing the maps of $|\nabla T|$ at 5 km and 100 km, these highs are likely to be produced by coalescence between small-scale anomalies. We observe, indeed, that most of such $|\nabla T|$ highs, such as above PN, BM and ABM, represent the long-wavelength component of sharp and well-defined anomaly patterns, which merge each other at high-altitude forming a regional-scale magnetic signal. In Figure 9 we analyze jointly the magnetic and total gradient anomalies in specific areas of the Central Europe (red boxes in Fig. 9a) to the end of localize regions where reverse magnetization is dominant. In the area of the Bohemian massif (Fig. 9b) we immediately note that the magnetic field map above BM shows a wide magnetic low (left) in correspondence of the high of the total gradient intensity (right). This feature is interesting because, as discussed above, reverse magnetization area can be easily identified when the magnetic low of a reverse $T$ anomaly is replaced by a $|\nabla T|$ high (see Fig. 5). We may so conclude that the joint analysis of $T$ and $|\nabla T|$ suggests the presence of reversely magnetized sources above BM, even though paleomagnetic analysis did not reveal here a significant reverse magnetization.

In Figure 9c, we performed the same type of joint analysis in the area of the Brabant Massif, Namur Basin and Ardennes Massif. Magnetic and paleomagnetic studies (e.g. Molina Garza and Zijderveld, 1996) pointed out Paleozoic carbonate rocks




of Belgium characterized by uniform reverse polarity, as consequence of regional re-magnetization events occurred during

**Figure 9. Joint interpretation of magnetic and total gradient anomalies in Central Europe (a) over the Bohemian Massif (b), the Anglo-Brabant Massif (c), and Pannonian basin (d).**





tectonic and orogenic activity at Late Carboniferous times. In the magnetic field map, we observe two intense magnetic lows, with no clear presence of their relative highs, which transform to two highs in the $|\nabla T|$ map. These results, combined with the paleomagnetic information, allow interpreting these sources as characterized by a strong remanent component of magnetization. Further paleomagnetic studies were carried out in central Europe by Pucher (1994), where the analysis of

magnetic anomalies in Germany and Belgium pointed out pyrrhotite as the main carrier of remanent magnetization and similar magnetic feature was expected in other Paleozoic rocks of the central Europe. Other studies have been carried out by Zegers et al. (2003) about Permian remagnetization of Devonian limestones in northern France and Belgium.

The third area considered is the Pannonian basin (Fig. 9d), which formed contemporary to the Tertiary evolution of the Carpathians and the Eastern Alps, based on a SW to W dipping subduction (Bielik et al., 2004). Here an extended magnetic

low, visible up to satellite altitudes, lies over Slovakia and Hungary (Fig. 9d, left). Magnetic anomaly modelling was carried out in this area (Taylor et al., 2005, Kis at al., 2011), using magnetic field measurements of the CHAMP satellite. In their model the authors assume a strong remanent magnetization for the source according to exsolution of the hematite-ilmenite minerals found in the upper crust of the Pannonian Basin. Looking at the total gradient intensity map (Fig. 9d, right) we may find again a good correlation between the magnetic low anomaly and the high in the $|\nabla T|$ map.

We conclude that, besides the evidence of a strong contrast in magnetization between SW and NE European regions (hypothesis B) the presence of specific areas where remanent magnetization is dominant, represent an additional contribution to the CEML anomaly at 100 km altitude.

## 5.3 Magnetic Field at 350 km.

At 350 km, small-scale structures cannot be identified anymore, since their signal vanishes at such a great distance. The total

gradient intensity map, indeed, reveals long-wavelength anomalies covering most of central Europe. Moreover, the $|\nabla T|$ high is now located along the TESZ region. In south-eastern Europe, weak highs appear as a consequence of the partial coalescence between the TESZ anomaly and the Adriatic anomaly, which still represent the main appreciable magnetic contribution of southern Europe at satellite-altitude. On the Paleozoic platform, instead, the coalescence effects observed at 100 km altitude (Fig. 9) is essentially dissolved. The $|\nabla T|$ map, indeed, shows a large gradient zone starting from the high amplitudes above

the TESZ and prolonging toward eastern France, similarly to the total gradient intensity of the synthetic model in Figure 7b. Hypothesis A), based on the coalescence of anomalies due to reverse magnetization at satellite-altitudes, should have its counterpart in $|\nabla T|$ highs in Central Europe, according to the synthetic case described in Figure 6. But $|\nabla T|$ is there characterized by negligible values, so denoting that the dominant phenomenon is the presence of weak magnetization. This may be further verified by testing the models by Taylor and Ravat (1995) and Pucher and Wonik (1998). The results of the synthetic test are

shown in Figure 10 where we considered the same magnetic bodies of Taylor and Ravat, 1995 (see their Fig. 6). Normal magnetization is assigned to the magnetic source to the north-east, while the body in central Europe is considered mostly reversely magnetized, as result of coalescence effect of upper-middle crust sources. The magnetic field calculated at 350 km altitude shows a clear and intense reverse magnetic anomaly along the TESZ structure. The map of the total gradient, instead,





shows two strong maxima taking place above both central Europe and EEC, revealing, once again, its ability in discriminating the effect of magnetic sources. Looking at this test and at the "contact" test shown in Figure 7, the joint analysis of the maps of the magnetic field (Fig. 4c) and of the total gradient intensity (Fig. 8c) lead us to conclude that the hypothesis B is the most reliable to describe the CEML anomaly.

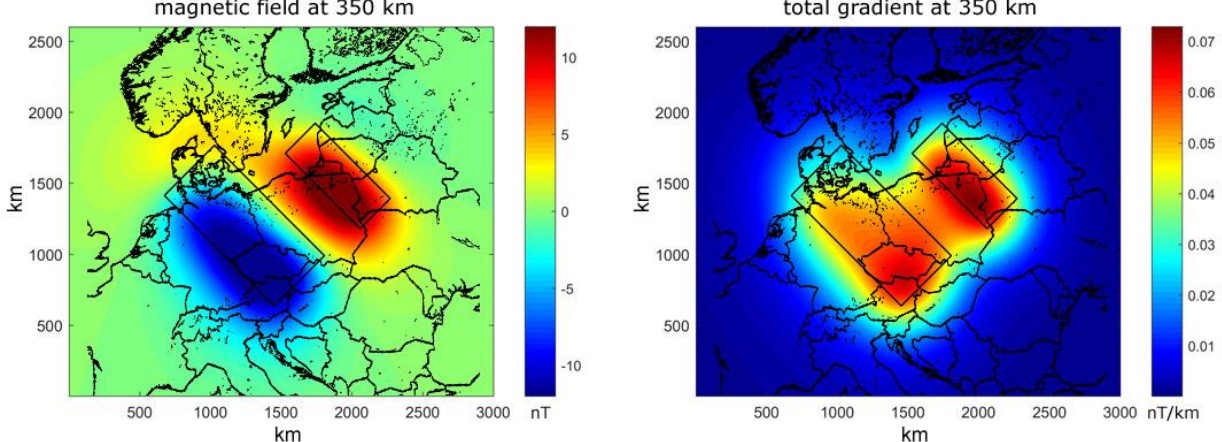

**Figure 10. Magnetic field (left) and Total Gradient intensity (right) from a synthetic model of central Europe in agreement with Taylor and Ravat (1995). This model refers to the hypothesis A described in this paper. Reverse magnetization is assigned to central Europe crust, and normal magnetization to the Precambrian platform. The magnetic field anomaly is computed at 350 km altitude (left). The total gradient intensity map (right) shows two highs above both the synthetic bodies.**

**Conclusions**

In this study we performed a step-by-step analysis of the European magnetic field, from low to high altitudes, aiming at explaining the regional-scale magnetic anomaly occurring in Central Europe. Previous interpretations were only conducted or at a regional scale, at high-satellite altitudes (say 350 km), or at a local scale, at low altitudes (say a few km).

10    In this study, instead, we performed a comparative study of the magnetic field and of its total gradient intensity at 5 km, 100 km and 350 km altitude. The joint analysis of $|\nabla T|$ and $T$ revealed to be a fundamental tool to identify the areas characterized by a weakly-magnetized crust and to recognize the presence of reversely magnetized sources. This task cannot be fulfilled analyzing the aeromagnetic field alone, due to the dipolar nature of the total magnetic field. The independence of the total gradient modulus from the directions of the inducing magnetic field and of the total magnetization vector appears clearly from

15    the result of a simple test involving both normally and reversely magnetized sources. We found that the sources are clearly





marked by highs of the total gradient intensity, while the magnetic field cannot help obtaining a similar information, due to the complex merging of the lows and highs of the several anomalies.

At large altitudes the long-wavelength magnetic anomalies are mainly related to the continental-scale magnetic properties of the crust. At 100 km altitude, the highs of the total gradient modulus of the aeromagnetic field well delineate the TESZ fault

line, marking out a magnetic boundary between two regions, which we interpreted as a contact area between the strongly magnetized middle-lower crust of the EEC (to the NE) and the weakly magnetized Paleozoic platform (to the SW). In particular, the total gradient intensity map was the main tool to favor the hypothesis of a scarce crustal magnetization in central Europe, in the area of the CEML, being there characterized by very low-amplitude values.

At 100 km altitude, the joint analysis of the total gradient intensity and of the magnetic field was also decisive in detecting

further magnetic sources in the Paleozoic platform, characterized by a reverse magnetization. In fact, highs of the total gradient intensity and strong lows in the magnetic field occur in the areas of the Brabant Massif, of the Pannonian basin and of the Bohemian Massif. This indicates the presence of sources with reverse magnetization, in agreement with rock magnetism surveys, which have shown the presence of minerals, such as pyrrhotite and magnetite, in metamorphic rocks and limestones of the Paleozoic formations.

The analysis of the total gradient intensity field at 350 km, instead, pointed out that the strong contrast in structural and magnetic properties is the main contribution to the magnetic field. $|\nabla T|$ highs, indeed, are well located above the TESZ region, with a trend directed toward the Paleozoic Platform. This is typical of a contact-like effect, as shown by synthetic modeling. On the other hand, there are no significant highs of the total gradient intensity in Central Europe, leading us to consider a weak to average magnetization for the Paleozoic crust.

In conclusion, the large-altitude magnetic low of Central Europe can be interpreted as being due mainly to a weakly magnetized thin crust *versus* a strongly magnetized and thick crust for the EEC, bounded by the TESZ. The presence of rocks with a strong remanent component of magnetization, characterizing some areas of the Paleozoic platform, cannot contribute to the formation of the whole CEML, and their effects vanish rapidly at satellite elevations.

**Data availability**

EMMP high-resolution dataset are available near Getech, UK, info@getech.com, MF7 model data can be downloaded at http://geomag.org/models/MF7.html.

**Author contribution**

M.M. and M.F. conceived of the presented work and with J.D.F. wrote the paper. M.M. performed the computations, derived the models and produced the figures. M.F. and J.D.F. supervised the findings of the work. All authors discussed the results

and contributed to the final manuscript.



**Competing interests**

The authors declare no competing interests.

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
