# Peer review of "Joint analysis of the magnetic field and Total Gradient Intensity in Central Europe"

_Solid Earth, 2019_

## Referee Comment (RC1) · Fabio Caratori Tontini (Referee) · 29 Mar 2019

I carefully read the paper by Milano et al. and I found it an interesting contribution which I recommend for publication. The authors aim to solve the controversy regarding the interpretation of large-scale anomalies in Central Europe, which will impact on other studies regarding orogeny and general tectonics. This topic is such is of broad relevance. I am attaching a list of minor points in an annotated manuscript for some further clarification.

Fabio Caratori Tontini

Please also note the supplement to this comment:

[Figure]

https://www.solid-earth-discuss.net/se-2019-40/se-2019-40-RC1-supplement.pdf

---

## Referee Comment (RC2) · Jörg Ebbing (Referee) · 15 Apr 2019

I generally like the paper and consider this study of broad interest as its addresses a controversy in the geomagnetic community. I suggest a few small changes to broaden its appeal as the authors sometime get lost in the (geological) details, but from the introduction I would have expected a more detailed discussion of the pros and cons of aeromagnetic and satellite models for study this region.

page 4, line 7: no ending for sentence

page 4, line 4: KTB was not a deep seismic profile. These were the DEKORP profiles. KTB was the deep drilling for which as well seimsic studies have been carried out.

page 4, line 20: Korja & Heikkinnen , 2005: This is a study on the deep (Svecofennian)

[Figure]

part of Europe, not the shallow one as stated here.

Figure 3: I would suggest to delete the decimal points and to use an even spaced colour scale.

page 8, line 20ff: Could you please add the depths at which the sources are placed.

page 9, line 16: At which depth are the 19 sources placed? And is the regional field a consequence of the orientation of the inducing field or how does t relate to the sources?

page 11, line 9: Maybe show an intermediate model with constant magnetisation to demonstrate the effect of geometry only

Section 5.1-5.3 I find this discussion to be a bit odd and lengthy. Your main discussion was the origin of the magnetic anomaly over the TESZ , so why here you add a very detailed discussion of (all) European anomalies? I think this part could be shorten for clarity and to increase the appeal of the paper for its readers.

page 20, line 20: What about differences between EMMP and MF7? I miss a more detailed discussion how the source geometry results in the field and a specific discussion of the spectral content of MF7 vs. EMMP. I think a lot of people use MF7 and here you could demonstrate its pros and cons in interpretation a large scale anomaly as observed in central Europe. I would prefer such a discussion in comparison to the discussion of local anomalies in the text.

Page 20, line 25: data are available from GETECH, not near GETECH

---

## Editor Comment (EC1) · Nicolas Gillet (Editor) · 15 Apr 2019

Dear Authors, Having read the comments by the referees, I consider your paper is appectable for publication with minor revisions, provided you address the points raised by the referees. Best Regards, Nicolas Gillet.

---

## Author Comment (AC1) · 24 Apr 2019

Dear Dr. Caratori Tontini, Thank you, on the behalf of the co-authors, for your feedback and the constructive comments on our manuscript. We modified the text according to your suggestions: - Page 2, line 14: we clarify why we used the MF7 magnetic model data instead of the EMMP dataset, at 350 km altitude. The reason is that we noted not negligible edge effects in the map performed by upward continuation of EMMP data at so a large distance.

- Page 4. Line 9: we have added the sentence 'Archean-Paleoproterozoic episodes of accretion and reworking (Gaàl and Gorbatschev, 1987) and covered by thin series of Phanerozoic rocks (Plant et al, 1998)' that was erroneously removed from the text, as

well as the citations.

- Page 8, line 20: we agree with your comment: the use of the total gradient technique itself does not provide information on magnetization direction. This information may be however obtained by comparing it with the magnetic field data at the same altitude, that is the main purpose of the manuscript. We fixed such concept.

- Page 11, line 5: thank you for your remark: we changed the related text, since the total gradient analysis, in such case of coalescence, does not help in discriminating the source effects one each other; it however provides information on the source regions with the stronger magnetization intensity.

- Page 12, line 6. We removed this sentence because unnecessary. The total gradient anomalies above the Precambrian Europe appear to be merged, because of a purely graphical artifact, due to the chosen colormap limits (maximum: 15 nT/km). We set this value to emphasize the weaker anomalies above the Paleozoic Europe.

- Page 14: our main goal is not addressed to a study of the TESZ anomaly only, we wanted instead to give a reasonable explanation of the CEML, so that we are interested to consider also the main magnetic provinces in central Europe, such as the Pannonian basin. Moreover, our analysis of the Pannonian basin (Figure 9D) was justified by a related objective of such analysis, shared also by previous researches on CEML anomaly: identifying the areas where reverse magnetization could be dominant.

- Page 16, line 7. We change from Moho (crust) to lithosphere as you suggested. However, in Milano et al. 2016, the Moho has been interpreted as the bottom of the magnetic crust thanks to the high agreement between the multiscale analysis results and the seismic data information.

Please read the attached modified version of the manuscript.

Please also note the supplement to this comment:

https://www.solid-earth-discuss.net/se-2019-40/se-2019-40-AC1-supplement.pdf

**Supplement:**

[revised manuscript text omitted]

---

## Author Comment (AC2) · 24 Apr 2019

Dear Prof. Ebbing thank you for your valuable comments and suggestions. Please note below the answers to the individual comments, the modified version of the manuscript and the attached figure: - page 4, line 7: no ending for sentence:

We have fixed this mistake, by adding part of the sentence that we erroneously deleted. See attached the new version of the manuscript.

- page 4, line 4: KTB was not a deep seismic profile. These were the DEKORP profiles. KTB was the deep drilling for which as well seismic studies have been carried out:

Thank you for noting this mistake: we corrected it. We have modified this part and the relative references according to your comments.

[Figure]

- Figure 3: I would suggest to delete the decimal points and to use an even spaced colour scale:

Thank you: Figure 3 was modified by using a linear color scale and removing the decimal point.

- page 8, line 20ff: Could you please add the depths at which the sources are placed:

We have added information about depth position of the model sources.

- page 9, line 16: At which depth are the 19 sources placed? And is the regional field a consequence of the orientation of the inducing field or how does t relate to the sources?:

You are right: the 19 sources are placed at different depths ranging between 2 and 10 km. We have added this information in text. The long-wavelength field observed at 350 km altitude is the effect of coalescence between the anomalies of the 19 sources. Total magnetization direction was: ; inducing field direction was: . So, we think that the shape of the coalesced anomaly is dominated by the remanent magnetization, whose intensity is stronger southern. Note that the total gradient intensity map shows clearly a more intense magnetization southern, in accordance with the values selected for the sources.

- page 11, line 9: Maybe show an intermediate model with constant magnetization to demonstrate the effect of geometry only

According to your suggestion, we have added a new model and inserted it in the supplementary material. In this model, the magnetic field and total gradient field of the TESZ model were calculated assuming for both the 'Paleozoic' and the 'Precambrian' synthetic crusts 2 A/m of magnetization. The magnetic field map at 350 km shows a very low-intensity anomaly, this time related to only the variation from the thin southwestern crust to the thicker one to the northeast. Such structure is also imaged in the total gradient map by an extended maximum amplitude along the contact

line. Therefore, the case of two magnetized crust differing exclusively in the structural features may contribute to the magnetic field but cannot explain completely such magnetic anomaly. The combination of both structural and magnetic property differences, instead, seems the best hypothesis to explain the observed magnetic field above Europe.

- Section 5.1-5.3 I find this discussion to be a bit odd and lengthy. Your main discussion was the origin of the magnetic anomaly over the TESZ , so why here you add a very detailed discussion of (all) European anomalies? I think this part could be shorten for clarity and to increase the appeal of the paper for its readers.

We have reduced the length of section 5.1, removing supplementary information on small-scale magnetic anomalies, that are superfluous and beyond the main topic of this study. However, we point out that our analysis is not exclusively focused on the TESZ area. The joint analysis is performed all over central Europe (say the extension of the CEML). The need for such a study is justified by observing that previous interpretations were based not only on TESZ anomaly but also to the magnetic contributions of sources in Central Europe other than the TESZ structure.

- page 20, line 20: What about differences between EMMP and MF7? I miss a more detailed discussion how the source geometry results in the field and a specific discussion of the spectral content of MF7 vs. EMMP. I think a lot of people use MF7 and here you could demonstrate its pros and cons in interpretation a large scale anomaly as observed in central Europe. I would prefer such a discussion in comparison to the discussion of local anomalies in the text.

The CHAMP MF7 crustal field was the best available satellite derived field at the time of the study. Due to the satellite's orbital height the wavelength resolution of this field was ∼150 km. Following Fletcher et al. (2011), the EMMP compilation used all original flight line data down to point ground data available from each county. So, the resolution of many surveys allowed an optimum grid to be generated down to 1km. Technical data

concerning these surveys also allowed the IGRF correction to be applied. With any compilation, merging surveys with a range of spatial survey sizes, ages, instruments, processing methods can and did generate small differences which will not necessarily average out over the compilation. Since the EMMP compilation generally lacked long wavelength control (i.e. $\lambda \geq \sim 150$ km) due to limited size of surveys, the final processing step, after gridding at 1km, was to drape it onto the CHAMP's MF7 crustal field (MF7, Maus, 2010).

- Page 20, line 25: data are available from GETECH, not near GETECH.

Thank you, we fixed it.

Please also note the supplement to this comment:
https://www.solid-earth-discuss.net/se-2019-40/se-2019-40-AC2-supplement.pdf

[Figure]

**Fig. 1.**

---

## Author Comment (AC3) · 25 Apr 2019

Dear Dr. Gillet

Thank you for the nice review editing, which we much appreciated. We have completed the responses to each reviewer comments, as you may find online.

Obviously, we modified the manuscript according to such comments, as you may verify in the attached file. Modifications are yellow-enlighted

Best Regards Maurizio Milano

Please also note the supplement to this comment:
https://www.solid-earth-discuss.net/se-2019-40/se-2019-40-AC3-supplement.pdf